# A Method for Rapid Polyethyleneimine-Based Purification of Bacteriophage-Expressed Proteins from Diluted Crude Lysates, Exemplified by Thermostable TP-84 Depolymerase

**DOI:** 10.3390/microorganisms11092340

**Published:** 2023-09-19

**Authors:** Beata Łubkowska, Edyta Czajkowska, Ireneusz Sobolewski, Natalia Krawczun, Agnieszka Żylicz-Stachula, Piotr M. Skowron

**Affiliations:** 1Faculty of Health and Life Sciences, Gdansk University of Physical Education and Sport, Gorskiego 1, 80-336 Gdansk, Poland; 2Department of Molecular Biotechnology, Faculty of Chemistry, University of Gdansk, Wita Stwosza 63, 80-308 Gdansk, Poland; edyta.czajkowska@ug.edu.pl (E.C.); ireneusz.sobolewski@ug.edu.pl (I.S.); natalia.krawczun@ug.edu.pl (N.K.); agnieszka.zylicz-stachula@ug.edu.pl (A.Ż.-S.); piotr.skowron@ug.edu.pl (P.M.S.)

**Keywords:** biotechnology, depolymerase, capsule, envelope, biofilm, biofilm removal, glycosylase, polyethyleneimine, TP-84, bacteriophage, polysaccharide, purification, thermophile, *Geobacillus stearothermophilus*

## Abstract

Purification of bacteriophage-expressed proteins poses methodological difficulties associated with the need to process entire culture medium volume upon bacteriophage-induced bacterial cell lysis. We have used novel capsule glycosylase-depolymerase (TP84_26 GD) from bacteriophage TP-84, infecting thermophilic *Geobacillus stearothermophilus* bacteria, as a representative enzyme to develop a method for rapid concentration and purification of the enzyme present in diluted crude host cell lysate. A novel variant of the polyethyleneimine (PEI)-based purification method was devised that offers a fast and effective approach for handling PEI-facilitated purification of bacteriophage-expressed native proteins. Due to the very basic nature of PEI, the method is suitable for proteins interacting with nucleic acids or acidic proteins, where either mixed PEI-DNA or RNA–protein complexes or PEI–acidic protein complexes are reversibly precipitated. (i) The method is of general use, applicable with minor modifications to a variety of bacteriophage cell lysates and proteins. (ii) In the example application, TP84_26 GD was highly purified (over 50%) in a single PEI step; subsequent chromatography yielded a homogeneous enzyme. (iii) The enzyme’s properties were examined, revealing the presence of three distinct forms of the TP84_26 GD. These forms included soluble, unbound proteins found in host cell lysate, as well as an integrated form within the TP-84 virion.

## 1. Introduction

The thermostable TP-84 TP84_26 GD is the enzyme which is produced by TP-84 bacteriophage-infected *G. stearothermophilus* [1,2,3,4] cells. It was purified in native form and characterized by us [3,5]. We have also cloned expressed TP84_26-coding gene into *Escherichia coli* and purified the recombinant variant TP84_26 GD. The apparent biological role of the enzyme is to make passages for the TP-84 thermophage through a very thick capsule of *G. stearothermophilus,* estimated to be tens of times larger in volume than the vegetative host’s cells. Bioinformatics analysis has revealed that TP84_26 GD belongs to glycosyl hydrolase family 18,l, exhibiting homologies to the spore germination protein YaaH (cell cycle control, cell division, chromosome partitioning) [5]. The enzyme forms a large tetramer of approx. 470 kDa apparent molecular weight. It was hypothesized that this comprises a mechanism for slowing down diffusion of the enzyme released from lysed host cells, allowing for initial pre-digestion of uninfected host cell capsules located in the neighborhood of infected cells. We have used this novel TP84_26 GD, derived from thermophilic bacteriophage TP-84, to develop a method for rapid concentration and purification of the native enzyme present in diluted crude host cell lysate. PEI is a commonly used polymer for initial stages of protein purification [6,7,8]. It is produced in either linear or branched form, composed of repeating units containing an amine group and two carbon aliphatic (CH2_2_)(CH_2_) spacers. The linear PEIs contain secondary amines, while the branched PEIs contain primary, secondary and tertiary amino groups [9,10]. In either form, PEIs are typically supplied as a high-molecular-weight-concentrated solution in water [10]. PEI stock is highly basic; thus, the working solutions are prepared by diluting the stock solution to the desired concentration (typically 10%) and neutralized with HCl to pH 7.5–8. PEI-mediated protein purification is meant as the first stage, operating on crude cell lysate. Thus far, it has been used in the following modes: (i) simple precipitation of nucleic acids and some acidic proteins in low-salt crude lysates (more than approx. 200 mM ionic strength, adjusted individually), as they form insoluble complexes with PEI [11]; (ii) the same procedure, but conducted in low ionic strength, suitable for use with neutral to basic proteins that do not interact with nucleic acids or PEI; and (iii) the most refined variant for nucleic acid-interacting proteins co-precipitated with them at low ionic strength with limited amounts of PEI, so as to avoid its ‘protein stripping’ activity from nucleic acids. Upon centrifugation of mixed complexes, a protein of interest is recovered by selective back-extraction with high-salt buffers. All these variants are conducted on highly concentrated ‘standard’ cell lysates, obtained without employing a bacteriophage, by various cell disintegration methods, such as sonication, lysozyme digestion and mechanical disruption, among others. Contrary to these three variants, we show here a novel variant applicable to highly diluted bacteriophage-induced cell lysates, comprising the entire original culture volume. The lysates obtained in this way are typically up to 100 times less concentrated as compared to the ‘standard’ cell lysates.

The progress of PEI-mediated protein purification can be monitored using various techniques, such as enzymatic assays or gel electrophoresis [12]. In general, PEI acts as a suitable carrier for the proteins, providing a favorable environment for their stability [13].

Proteins can be recovered from the PEI solution using standard protein purification techniques. This may involve steps such as centrifugation, filtration, chromatography, gel filtration or precipitation, depending on the specific requirements of the target protein. Besides scientific applications, this study will be useful for scaled up production biotechnology, as the novel variant offers great cost and time savings, bypassing the necessity to process large volumes of bacteriophage-generated host cell lysates.

## 2. Materials and Methods

### 2.1. Growth Conditions of the TP-84 Bacteriophage Host and the Bacteriophage

The *G. stearothermophilus* strain 10 and TP-84 bacteriophage used in the study were initially acquired from Epstein and Campbell [1]. A streptomycin-resistant mutant of *G. stearothermophilus* 10 was created by serial passaging on solid TYM media with increasing concentrations of streptomycin. The concentrations used were: 5 μg/mL, 10 μg/mL, 20 μg/mL and 50 μg/mL. This was achieved by using 300 μL of dense, overnight (17 h) culture (approximately 10^8^ cells) of *G. stearothermophilus* 10. For preparative-scale growth, bacteria were cultured with vigorous aeration at 55 °C. The TYM-rich medium [2] was used, which contained calcium and magnesium ions, as well as fructose, known to stimulate TP-84 yield. The TYM medium composition included: Pepton K: 20 g/L (from BTL, Warsaw, Poland), yeast extract: 4 g/L (from BTL), CaCl_2_, MgCl_2_, fructose, and streptomycin were added after sterilization/cooling to final concentrations of 5 mM, 10 mM, 0.5%, and 50 μg/mL, respectively. For solid media, the bottom TYM agar contained 2% agar, while the top TYM agar contained 0.6%. To prepare the TP-84/*G. stearothermophilus* 10 str^R^ cell lysate as starting material for TP84_26 GD production, the strain was grown at 55 °C overnight. Then, 5 mL of the overnight culture was used to inoculate 1 L of TYM medium. The culture was incubated at 55 °C with vigorous aeration for approximately 4 h until reaching an OD_600nm_ of approximately 0.4. Fructose was added again to a final concentration of 0.5%, and the culture was continued for an additional 30 min. The liquid culture was then infected with 6.2  ×  10^10^ TP-84 particles at a multiplicity of infection (M.O.I.) of approximately 10.

The infection was allowed to proceed for 10 min without shaking. After the infection, the culture was incubated at 55 °C with vigorous aeration for 7.5 h. When the OD_600nm_ dropped to 0.1, a stock solution of phenylmethylsulfonyl fluoride (PMSF) in isopropanol (2.5 mL, 0.2 M) was added to a final concentration of 0.5 mM. Additionally, 3 mL of chloroform was added to the culture. The mixture was then left overnight at room temperature to complete cell lysis. The cell debris was separated by centrifugation, and the supernatant was collected. The supernatant was supplemented with 20 mM EDTA and 80 mM NaCl to remove weakly bound proteins from PEI complexes, formed during subsequent enzyme purification.

### 2.2. Protein Standards, DNA Markers, Protease Inhibitor and Other Chemicals

For the protein standards, we obtained the following from GE Healthcare (Chicago, IL, USA): PageRuler™ Plus Prestained Protein Ladder (catalog number 26619), LMW SDS Marker Kit (catalog number 17044601), and Coomassie™ Brilliant Blue (catalog number 20279). The DNA markers were acquired from Thermo Fisher Scientific (Waltham, MA, USA) and GE Healthcare Bio-Sciences AB (Piscataway, NJ, USA). The protease inhibitor used in the study was Protease Inhibitor Cocktail Tablets, EDTA-Free (catalog number S8820), obtained from Sigma-Aldrich (St. Louis, MO, USA). Sephacryl S300 HR and all other necessary chemicals were also acquired from Sigma-Aldrich (St. Louis, MO, USA).

### 2.3. Chromatographic Media and Centrifugal Devices

The chromatographic resins used in the study were obtained from GE Healthcare Bio-Sciences AB. Specifically, we utilized the following resins: Q-Sepharose™ Fast Flow (catalog number 17–0510-01), Blue-agarose™ Fast Flow (catalog number 17–0948-01), and Heparin-Sepharose™ 6 Fast Flow (catalog number 17–0998-01). The centrifugal devices used in the study were sourced from Sartorius Stedim Biotech GmbH (Göttingen, Germany). Specifically, we utilized the following devices: SIGMAFAST™ Vivaspin Turbo 15 PES with a molecular weight cutoff (MWCO) of 10,000 (catalog number VS15T02) and VivaSpin^®^ Turbo 15 RC with a cutoff of 100 kDa (catalog number VS15T41).

## 3. Results

### 3.1. Purification of the Native TP84_26 GD Employing the Novel Method for Protein Purification Using PEI

Purification of the TP84_26 GD [5] consisted of the following steps: (i) Co-precipitation: TP84_26 GD, some acidic proteins, nucleic acids and their complexes with proteins were co-precipitated using a limited concentration of PEI applied directly to the TP-84/*G. stearothermophilus* 10 str^R^ lysates. Due to low concentration of proteins present in the whole culture volume, 1% PEI buffered solution was used as opposed to commonly used 10% solutions. This was followed by selective back-extraction with high salt buffer to isolate the TP84_26 GD. Even though TP84_26 GD is present in minute amounts in TP-84/*G. stearothermophilus* 10 str^R^ lysates, this single co-precipitation step has still resulted in the preparation containing approx. 50% of the TP84_26 GD. Further, standard chromatographic techniques were used to obtain the final, homogeneous preparation. This step is followed by: (ii) phosphocellulose P11 chromatography (negative step, binding impurities); (iii) Q-Sepharose chromatography; (iv) heparin-agarose chromatography; and (v) size fractionation and concentration using Vivaspin membrane-spin devices with a 100 kDa cut-off. This step helped to remove contaminants smaller than the TP84_26 GD (112 kDa) (Figure 1).

#### 3.1.1. Step 1: Fractionation of TP-84/*G. Stearothermophilus* 10 strR Lysates Using PEI


**
*Step 1A: titration of TP-84/G. stearothermophilus 10 strR lysates with PEI.*
**


Clear lysate samples (5 mL) were titrated with 1% PEI solution (pH 7.5, adjusted with HCl as 10% stock) using varying volumes (μL) of 5, 10, 20, 35, 50, 70, 100, 150, 200 and 300. The goal was to determine the amount of PEI that generated the most intense turbidity, without exceeding that optimal PEI concentration. The samples were shaken briefly for 5 s and then incubated for 1 h at 4 °C. The optical density at 600 nm (OD_600nm_) was measured (Figure 2A,B). For the actual TP84_26 GD purification, the previously determined optimal (i.e., the lowest to cause the highest turbidity) PEI concentration was used. This is very important not to exceed the titration-determined PEI concentration, as its excess tends to outcompete the DNA binding proteins, causing their release from formed complexes. Furthermore, as seen in Sample 10, the excess of PEI reduces turbidity, thus forming less precipitate. This is the result of both the release of proteins and the partial solubilization of nucleic acids–PEI complexes, caused by dominating PEI charge.


**
*Step 1B: Scaled up co-precipitation of the TP84_26 GD with PEI.*
**


The determined optimal volume of 1% PEI solution was proportionally multiplied (e.g., 1000/5 = 200 times) for the use for 1 L TP-84/*G. stearothermophilus* 10 str^R^ cleared lysate. Then, 40 mL of 1% PEI solution was added with rapid mixing to 1 L of TP-84/*G. stearothermophilus* 10 str^R^ cleared lysate, the mixture was stirred for 1 h at 4 °C and then centrifuged for 20 min at 6000×  *g*. This step aimed to remove PEI complexes with nucleic acids, nucleic acid–protein complexes and acidic proteins. The supernatant was diluted 1:1 with distilled water and centrifuged again to collect any remaining complexes. The centrifuged complexes from both steps were combined.


**
*Step 1C: Back-extraction of proteins from PEI complexes with high salt buffer.*
**


The combined complexes were resuspended in 20 mL of buffer E containing Tris-HCl (30 mM, pH 8.0 at 20 °C), EDTA (0.5 mM), NaCl (20 mM), Triton X-100 (0.01%), Tween 20 (0.01%), glycerol (5%), β-mercaptoethanol (5 mM), and PMSF (0.1 mM). The resuspension was incubated overnight at 4 °C with gentle stirring. This step was important for high enzyme recovery, as the PEI complexes pellet is very sticky. Further, high salt buffer (6.75 mL of 0.5 M NaCl in E buffer) was added to the resuspension to final concentration of 220 mM. The mixture was stirred for 15 min and then centrifuged at 6000×  *g* for 20 min at 4 °C. The resulting supernatant (30 mL) was dialyzed against DEAE buffer (Figure 3A).


**
*Step 1D (alternative): Back-extraction of proteins from PEI complexes with low pH/high salt buffer.*
**


As an alternative back-extraction buffer variant, a low pH phosphate-based buffer (50 mM K/PO_4_, pH 6.0, 0.5 M NaCl) was used for direct resuspension of PEI complex pellets overnight. Further, the mixture was stirred for 15 min and then centrifuged at 6000×  *g* for 20 min at 4 °C.

#### 3.1.2. Step 2: Phosphocellulose P11 Chromatography

Following the PEI fractionation and dialysis steps, the supernatant was subjected to phosphocellulose P11 chromatography. The dialyzed supernatant was loaded onto a column (1.8 mL) packed with phosphocellulose P11 resin. The column was pre-equilibrated with E buffer at pH 8.0. This chromatographic step acted as a negative selection, with the TP84_26 GD passing through the column in the flow-through fraction. The collected TP84_26 GD preparation was subsequently dialyzed against E buffer containing 100 mM NaCl to ensure the enzyme-stabilizing buffer conditions (Figure 3A).

#### 3.1.3. Step 3: Q-Sepharose Chromatography

After the phosphocellulose P11 chromatography step, the dialyzed TP84_26 GD solution was further purified using Q-Sepharose chromatography. The TP84_26 GD solution was loaded onto a Q-Sepharose column (1.8 mL) that had been equilibrated with E buffer supplemented with 100 mM NaCl. The column was then washed with 10 mL of E buffer to remove any non-specifically bound proteins. Elution of the TP84_26 GD was achieved by employing a step-wise increase in NaCl concentration. Two steps were performed—first using E buffer containing 250 mM NaCl, followed by E buffer containing 500 mM NaCl.

Fractions collected were analyzed using 10% SDS-PAGE to determine the presence of TP84_26 GD. The fractions containing the highest amounts of TP84_26 GD (4 mL) were pooled together and diluted with 20 mL of E buffer for further processing and characterization (Figure 3A).

#### 3.1.4. Step 4: Heparin-Agarose Chromatography

The diluted fractions containing the TP84_26 GD from the previous purification step were loaded onto a heparin-agarose column (2.5 mL) that had been pre-equilibrated with E buffer. The heparin-agarose column acted as a negative selection, with the TP84_26 GD eluting in the flow-through fractions (Figure 3A).

#### 3.1.5. Step 5: Membrane Size Fractionation and Concentration

The TP84_26 GD preparation (23 mL) was concentrated and simultaneously used to remove small proteins through centrifugation in a VivaSpin^®^ Turbo 15 RC device with a 100 kDa cutoff, resulting in a final volume of 6 mL (Figure 3B). The concentrated TP84_26 GD solution was then diluted with E buffer and concentrated again, yielding approximately 1 mL of final solution. This solution was dialyzed against a storage buffer S (50 mM Tris-HCl pH 8.0 at 23 °C, 100 mM NaCl, 2.5 mM MgCl_2_, 2.5 mM CaCl_2_, 0.01% Triton X-100, 0.50% glycerol). To the obtained 0.6 mL of the final TP84_26 GD preparation at the enzyme concentration of 2.0 mg/mL, a reducing agent TCEP was added to a final concentration of 1 mM.

### 3.2. General Protocol for PEI-Mediated Purification of Proteins from a Bacteriophage/Host Cell Lysate


**1. Bacteriophage culturing.**


Grow and infect a bacteriophage host under conditions favorable for maximum bacteriophage burst to stimulate the biosynthesis of the bacteriophage protein of interest. Add chloroform to 0.1% (except cultures of lipid membrane-enveloped bacteriophages), shake for 10 min and allow host cell lysis to complete. Centrifuge and discard cell debris.

***Note 1:**** It is recommended to repeat centrifugation twice to minimize carry over of debris and reduce amounts of contaminating proteins*.


**2. Titration of a bacteriophage/host cell crude lysate with PEI.**


Aliquot 5 mL crude lysate samples cooled to 4 °C and add, with rapid mixing for 5 s, 1% PEI solution buffered with HCl (pH 7.5). Incubate for 1 h at 4 °C. Measure OD_600nm_ and select the sample with the highest one for calculations of scaled up PEI co-precipitation.

***Note 2:*** *This is the critical step which determines optimal PEI concentration for co-precipitation of mixed complexes with nucleic acids, nucleic acid-bound proteins and some other proteins, usually those of low pI. Adding an insufficient amount of PEI will result in incomplete recovery of the protein of interest from crude lysates, while adding too much PEI will result in outcompeting nucleic acid-interacting proteins and their release from the co-precipitate. Further, partial solubilization of the co-precipitate may occur.*

***Note 3:*** *As culture media may vary greatly in composition and thus also in ionic strength and pH, every case needs individual adjustments. Depending on the affinity of the protein of interest to nucleic acids or to PEI, the clarified crude lysate may need adjustments by changing pH and salt concentration (dilution or addition of NaCl). In some cases (e.g., for the TP84_26 GD), the yields can be increased by diluting the clarified supernatant in water after the PEI co-precipitate centrifugation. A decrease of ionic strength will result in the occurrence of additional turbidity, caused by the formation of residual PEI complexes. Upon centrifugation, the co-precipitates are to be combined*.


**3. Scaled up PEI complex precipitation.**


Multiply the volume of 1% PEI solution needed for precipitation of the complexes from the entire cleared crude lysate volume by a factor obtained by dividing the lysate volume (in mL) by five. Add 1% PEI solution slowly with rapid stirring of the lysate at 4 °C. Continue stirring at moderate speeds for 1 h. Centrifuge for 20 min at 6000× *g*, discard supernatant, aspirate any remaining traces of the supernatant and discard.


**4. Back-extraction of the protein of interest from PEI complexes.**


Thoroughly resuspend the co-precipitate in the initial buffer (at approx. 1/30–50 proportion of the original cleared lysate volume) for 1 h to overnight. A good starting composition is that of buffer E (30 mM Tris-HCl (pH 8.0 at 20 °C), 0.5 mM EDTA, 20 mM NaCl, 0.01% Triton X-100, 0.01% Tween 20, 5% glycerol, 5 mM βME, 0.1 mM PMSF). Add NaCl from 0.5–1 M stock solution to a final concentration of 200 mM. Stir for 30 min, centrifuge for 20 min at 6000×  *g* and resuspend in buffer E, supplemented with final 400 mM for 1 h to overnight. Take 100 mL aliquots from every step (including cleared lysate), centrifuge them briefly for 5 min and monitor the supernatants on SDS/PAGE for progress in the release of a protein of interest. Repeat the resuspension cycles with increasing NaCl and/or decreasing pH, until a maximum amount of protein of interest is recovered.


**
*Note 4:*
**
* Since proteins of interest will vary greatly in their affinity toward nucleic acids and/or PEI, every case needs to be fine-tuned individually to determine the best salt/pH conditions for back-extraction. Once determined, use just the conditions sufficient to eliminate the need for sequential back-extractions. After the procedure is completed, the preparation is ready for subsequent purification using standard techniques.*


### 3.3. Method Validation

The validation of the method described in this study has been successfully conducted and reported by Łubkowska et al. [5]. 

## 4. Discussion

During our TP-84 bacteriophage studies, we needed to confirm the biosynthesis and the bioinformatically detected function of TP84_26 GD. To achieve those purposes, it had to be purified. However, there was a major obstacle—the necessity to process large volumes of the enzyme solution that is naturally produced in low amounts. Since bacteriophage proteins are, as a rule, initially obtainable as very dilute solutions, corresponding to starting culture volumes, we have developed a novel strategy to purify bacteriophage proteins. It employs PEI precipitation directly from TP-84-infected *G. stearothermophilus* 10 str^R^ lysates, while preserving the protein’s enzymatic activity. As PEI is an extended aliphatic chain polymer with charged amino and imino groups, its opposite charge and charged group spacing is, in a chemical sense, complementary to nucleic acids, which are linear polymers with regularly spaced negative charges. Thus, PEI is uniquely suited to precipitate DNA, RNA (and acidic proteins) as insoluble, but reversible salts. In general, PEI, being highly hydrophilic and possessing amino and imino groups prone to ionic and hydrogen bond formation with side chains of proteins, is not harmful to proteins or even produces a stabilizing effect. Thus, it is used in purification protocols to remove DNA, RNA and acidic proteins from cellular lysates. So far, it has been used in two modes, described in detail in the Introduction: (i) in coprecipitation of low salt buffers, and (ii) in medium-to-high salt buffers for the removal of DNA and RNA stripped of interacting protein. Contrary to these modes, this paper presents a universally advantageous novelty in PEI application. In this mode (iii), TP84_26 GD was successfully isolated and concentrated from large volumes of the *G. stearothermophilus* lysates using PEI. Those steps were followed by chromatography, yielding homogeneous TP84_26 GD protein. After the PEI stage alone, with efficiency surpassing typical results at the chromatographic stage, the preparation contained a very high proportion of the TP84_26 GD protein—approx. 50%. We believe that the proposed application of PEI to dilute protein solutions may possibly find usages far wider than protein isolation from bacteriophage lysates and contribute to scaled-up applications in scientific and industrial processes.

## 5. Conclusions

(1)The developed novel method for purification of TP84_26 GD, utilizing the polyethyleneimine (PEI)-mediated fractionation and chromatographic techniques, has proven to be effective in isolating and purifying the enzyme. The protocol successfully removed unwanted contaminants and allowed for the selective concentration and recovery of TP84_26 GD forms from the TP-84/*G. stearothermophilus* lysates.(2)The purification scheme involved the following steps: co-precipitation with PEI, chromatography on phosphocellulose P11, Q-Sepharose, and heparin-agarose media, and size fractionation through centrifugation using Vivaspin Turbo 15 RC devices. The obtained TP84_26 GD preparations exhibited high purity, as confirmed by SDS-PAGE analysis.(3)The validity of the developed methods was supported by their successful application in TP84_26 GD purification and demonstrated reproducibility. The purified TP84_26 GD samples, obtained using these protocols, can be further utilized in various applications, such as enzymatic studies, biotechnological applications or therapeutic purposes.(4)The presented purification methods offer efficient and reliable strategies for obtaining highly pure TP84_26 GD, contributing to the understanding and utilization of this enzyme in various fields.

## Figures and Tables

**Figure 1 microorganisms-11-02340-f001:**

Schematic diagram of the method for rapid PEI-based purification of bacteriophage-expressed proteins from diluted crude lysates.

**Figure 2 microorganisms-11-02340-f002:**
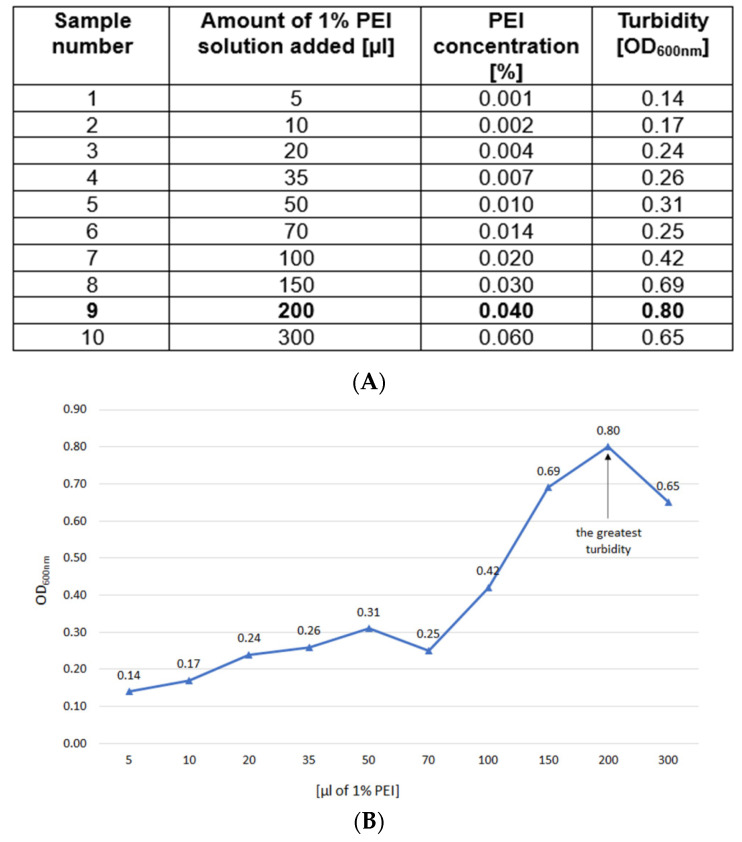
(**A**). TP-84/*G. stearothermophilus* 10 str^R^ cell lysates titration with 1% buffered PEI solution. Marked bold: the amount of 1% PEI solution selected for the E-D purification. (**B**). Titration curve as based on (**A**).

**Figure 3 microorganisms-11-02340-f003:**
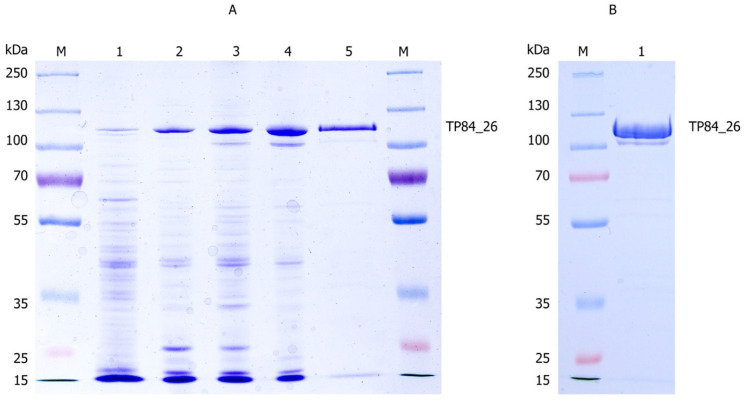
The purification process of the TP84_26 GD and other soluble TP-84 proteins from TP-84/*G. stearothermophilus* str^R^ lysates. (**A**) The 10% SDS-PAGE gel of the purification stages: Lane M, the molecular weight marker (PageRuler™ Plus Prestained Protein Ladder, ranging from 10 to 250 kDa). Lane 1, the TP-84_26 protein precipitated using TCA from 350 μL of lysate. Lane 2, the TP84_26 GD back-extracted from PEI complexes. Lane 3, the flow-through from the Phosphocellulose P11 column. Lane 4, the combined peak fractions from the Q-Sepharose chromatography. Lane 5, the flow-through from the Heparin-Sepharose column. (**B**) The final size fractionation and concentration of the TP84_26 GD. Lane M, the molecular weight marker (PageRuler™ Plus Prestained Protein Ladder, ranging from 10 to 250 kDa). Lane 1, TP84_26 GD after concentration using VivaSpin^®^ Turbo 15 RC. To identify the TP84_26 GD, protein bands were excised from the gel and analyzed using LC-MS. The bands corresponding to the TP84_26 GD were identified and marked with arrows.

## Data Availability

The authors confirm that the data supporting the findings of this study are available within the article.

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
