# Peer review of "A Method for Rapid Polyethyleneimine-Based Purification of Bacteriophage-Expressed Proteins from Diluted Crude Lysates, Exemplified by Thermostable TP-84 Depolymerase"

_microorganisms, 2023, doi:10.3390/microorganisms11092340_

Round 1

Reviewer 1 Report

Based on their previous work, the authors summarized and developed a protocol for the purification of proteins, here using a bacteriophage-expressed protein, the TP-84 capsule depolymerase as the model. In this work, the authors presented the purification of an enzyme with PEI precipitation. But there are some issues which should be improved before publication.

1.     As the authors claimed in the manuscript, the method could be used for the proteins interacting with nucleic acids or acidic proteins. But there is only one protein used in the method, more target protein should be used to test the method.

2.     Nucleic acids could be easily removed from the lysate, the results regarding comparison could be included.

3.     Some detailed comments:

Line 35-36, rewrite the sentence.

Line 48, start a new paragraph when introducing PEI.

Line 51, add brackets for CH2.

Line 73, remove the redundant commas. Line 319, remove the space between I and n.

There are some spelling and grammar errors, the authors should check the manuscript carefully to improve the readability.

Author Response

Dear Reviewer 1,

I appreciate your review of my protocol, as it significantly enhanced the quality of my manuscript. I have addressed your comments and suggestions below. Thank you for your valuable input.

Comments and Suggestions for Authors
Based on their previous work, the authors summarized and developed a protocol for the purification of proteins, here using a bacteriophage-expressed protein, the TP-84 capsule depolymerase as the model. In this work, the authors presented the purification of an enzyme with PEI precipitation. But there are some issues which should be improved before publication.
1.     As the authors claimed in the manuscript, the method could be used for the proteins interacting with nucleic acids or acidic proteins. But there is only one protein used in the method, more target protein should be used to test the method.
That is true, that we provided just a single example. However, PEI technique in its ‘classic’ form is known for many years and it was proven succesful on hundreds or more proteins purifications. Our novel PEI application variant is essentially based on the same principle. Thus it is safe to assume that it will work on a variety of proteins, present in diluted a bacteriophage lysates.

2.     Nucleic acids could be easily removed from the lysate, the results regarding comparison could be included.
That is true. However, the intention of the method developed was not to remove nucleic acids, but to form co-precipitate with a protein of interest and further back-extraction to achieve high purification factor.

3.     Some detailed comments:
Line 35-36, rewrite the sentence.
Line 48, start a new paragraph when introducing PEI.
Line 51, add brackets for CH2.
Line 73, remove the redundant commas. Line 319, remove the space between I and n.

Corrected

Comments on the Quality of English Language
There are some spelling and grammar errors, the authors should check the manuscript carefully to improve the readability.

The text was given for improvement to professional English correctors

Reviewer 2 Report

This manuscript by Lubkowska et al. describes the use of PEI (polyethyleneimine) to concentrate the bacteriophage capsular polymerase protein TP-84 from the cell lysate. The goal was to present a quick and simple method that would not require extensive prior concentration from the large volume of the lysate. The methods are detailed and seem to have achieved the goal. This is primarily a "Methods" paper, some portions written like a bench protocol. Thus, it may be useful to others for proteins with similar physicochemical properties. However, as the authors state and demonstrate, optimization of pH and PEI concentration will be required for each protein.

The main weaknesses are summarized below. However, very poor English has sometimes made it difficult to know if the strange meaning is due to bad science or just wrong English.

(1) In the Abstract, the protein is referred to as "a model enzyme". This is misleading. This polymerase, like all proteins, has its own special properties, such as its multisubunit structure, pH requirement for PEI complex formation, etc, which the authors had to actually optimize. In fact, the authors call it a "novel" capsule depolymerase (Line 46). A model enzyme cannot be also novel. Perhaps they meant that it is a "prototype" or "representative" polymerase of this class.

(2) Figure 1 should be called "Schematic diagram", not "Visual presentation". In this legend "Rapid" should be "rapid". This is not lack of English knowledge, but just carelessness.

(3) Line 156: What is "Optimal reagent concentration"? What is "Protocol 1" mentioned in Line 242 and elsewhere. It is not described anywhere.

(4) The authors have been studying this enzyme for a while, and this is a continuation and extension of some of their recent work, particularly Ref. 5. In fact, the authors have listed that paper  under the subheading "Related research article", which is somewhat unconventional. Fig. 2 in this manuscript is not just related, but essentially identical to that of Fig. 6 in Ref. 5, where the data were presented as a Bar graph.

 PEI as well as its cousin, polyethylene glycol (PEG), have been used for many years due to their amphiphilic properties. PEG and NaCl is a classical combination for macromolecule precipitation in molecular virology. The authors should emphasize on the novelty of this manuscript.

(5) If Fig. 2 must be presented, this could be an opportunity for the authors to explain why the solubility of TP-84 decreases from 200 to 300 microL. Moreover, instead of microL of PEI added, the authors should calculate the final concentration of PEI and present them in Fig. 2, both Panels A and B.

(6) Lastly, English needs improvement and correction almost in every sentence, some of which are just due to carelessness. For example "orgel electrophoresis" (Line 73).

Poor, and some places appear to be written carelessly.

Author Response

Dear Reviewer 2,   I appreciate your review of my protocol, which played a crucial role in enhancing and refining my manuscript. I have addressed your comments as outlined below. Thank you for your valuable input.     Comments and Suggestions for Authors This manuscript by Lubkowska et al. describes the use of PEI (polyethyleneimine) to concentrate the bacteriophage capsular polymerase protein TP-84 from the cell lysate. The goal was to present a quick and simple method that would not require extensive prior concentration from the large volume of the lysate. The methods are detailed and seem to have achieved the goal. This is primarily a "Methods" paper, some portions written like a bench protocol. Thus, it may be useful to others for proteins with similar physicochemical properties. However, as the authors state and demonstrate, optimization of pH and PEI concentration will be required for each protein. The main weaknesses are summarized below. However, very poor English has sometimes made it difficult to know if the strange meaning is due to bad science or just wrong English.   The text was given for improvement to professional English correctors   (1) In the Abstract, the protein is referred to as "a model enzyme". This is misleading. This polymerase, like all proteins, has its own special properties, such as its multisubunit structure, pH requirement for PEI complex formation, etc, which the authors had to actually optimize. In fact, the authors call it a "novel" capsule depolymerase (Line 46). A model enzyme cannot be also novel. Perhaps they meant that it is a "prototype" or "representative" polymerase of this class.   corrected   (2) Figure 1 should be called "Schematic diagram", not "Visual presentation". In this legend "Rapid" should be "rapid". This is not lack of English knowledge, but just carelessness.   corrected   (3) Line 156: What is "Optimal reagent concentration"? What is "Protocol 1" mentioned in Line 242 and elsewhere. It is not described anywhere. Actually we developed 2 protocols, but included here just one, that’s why we made a mistake.   Corrected   (4) The authors have been studying this enzyme for a while, and this is a continuation and extension of some of their recent work, particularly Ref. 5. In fact, the authors have listed that paper  under the subheading "Related research article", which is somewhat unconventional. Fig. 2 in this manuscript is not just related, but essentially identical to that of Fig. 6 in Ref. 5, where the data were presented as a Bar graph.   Yes, that was bit unconventional. We just followed an interesting concept of a new journal – MethodX (Elsevier) which encouraged to published ‘an method essence’ in the form similar to protocol as based on already published major paper by the same authors. We believe that such concept is of practical usefulness, as many readers are interested only in a new method, rather that reading more detailed studies. Nevertheless, we have removed this mini section "Related research article".    PEI as well as its cousin, polyethylene glycol (PEG), have been used for many years due to their amphiphilic properties. PEG and NaCl is a classical combination for macromolecule precipitation in molecular virology. The authors should emphasize on the novelty of this manuscript.   Done. It was already pointed out, but as requested, we emphasized that in other the manuscript locations   (5) If Fig. 2 must be presented, this could be an opportunity for the authors to explain why the solubility of TP-84 decreases from 200 to 300 microL. Moreover, instead of microL of PEI added, the authors should calculate the final concentration of PEI and present them in Fig. 2, both Panels A and B.   Yes, we believe Fig. 2 is useful for the readers for practical protocol follow up. Explained.   (6) Lastly, English needs improvement and correction almost in every sentence, some of which are just due to carelessness. For example "orgel electrophoresis" (Line 73). Comments on the Quality of English Language Poor, and some places appear to be written carelessly.   The text was given for improvement to professional English correctors

Round 2

Reviewer 1 Report

Compared with the original manuscript, the revised version is much better. However, the authors claimed in the responses that PEI co-precipitation is a general or classical method and they would not provide more examples. If so, please cite some classical references or examples to prove your viewpoints which is convenient for readers.

Author Response

Reviewer 1.

3 more ‘classic’  PEI usage references were added.

Reviewer 2 Report

The English is corrected. I believe the authors have responded to all my critiques, but it is difficult to read the improved areas since the Track Changes have been left on, and the only format submitted is PDF. This probably why "TP84" everywhere shows as "TP84 25 GD" ?! Generally, most authors submit two versions, one showing the Track Changes, and another, Clean Copy.  Minor typographical errors may have remained; for example CRediT (Line 400 in the Track Changes ON version), no space in theTECH..(Line 408); I don't know if there are others. Authors should thoroughly read the whole paper again, just to make sure. Anyway, the manuscript is now much better.

Author Response

Reviewer 2.

Indeed, our pdf conversion somehow was problematic. We have again reviewed the manuscript and corrected some minor typographic error. Now we are resubmitting the manuscript in the Word file only.
